# Explaining the Elusive Nature of a Well-Defined Threshold for Blood Transfusion in Advanced Epithelial Ovarian Cancer Cytoreductive Surgery

**DOI:** 10.3390/diagnostics14010094

**Published:** 2023-12-30

**Authors:** Alexandros Laios, Evangelos Kalampokis, Marios-Evangelos Mamalis, Amudha Thangavelu, Yong Sheng Tan, Richard Hutson, Sarika Munot, Tim Broadhead, David Nugent, Georgios Theophilou, Robert-Edward Jackson, Diederick De Jong

**Affiliations:** 1Department of Gynaecologic Oncology, St James’s University Hospital, Leeds LS9 7TF, UK; amudhathangavelu@nhs.net (A.T.); yong.tan@nhs.net (Y.S.T.); richard.hutson@nhs.net (R.H.); s.munot@nhs.net (S.M.); tim.broadhead@nhs.net (T.B.); david.nugent@nhs.net (D.N.); georgios.theophilou@nhs.net (G.T.); diederick.dejong@nhs.net (D.D.J.); 2Department of Business Administration, University of Macedonia, 54636 Thessaloniki, Greece; ekal@uom.edu.gr (E.K.); bad22019@uom.edu.gr (M.-E.M.); 3Center for Research & Technology HELLAS (CERTH), 6th km Charilaou-Thermi Rd, 57001 Thessaloniki, Greece; 4Department of Anaesthesia, St James’s University Hospital, Leeds LS9 7TF, UK; robert.jackson3@nhs.net

**Keywords:** epithelial ovarian cancer, complete cytoreduction, estimated blood loss, estimated blood volume, blood transfusion, intra-operative mapping, machine learning, explainable artificial intelligence

## Abstract

There is no well-defined threshold for intra-operative blood transfusion (BT) in advanced epithelial ovarian cancer (EOC) surgery. To address this, we devised a Machine Learning (ML)-driven prediction algorithm aimed at prompting and elucidating a communication alert for BT based on anticipated peri-operative events independent of existing BT policies. We analyzed data from 403 EOC patients who underwent cytoreductive surgery between 2014 and 2019. The estimated blood volume (EBV), calculated using the formula EBV = weight × 80, served for setting a 10% EBV threshold for individual intervention. Based on known estimated blood loss (EBL), we identified two distinct groups. The Receiver operating characteristic (ROC) curves revealed satisfactory results for predicting events above the established threshold (AUC 0.823, 95% CI 0.76–0.88). Operative time (OT) was the most significant factor influencing predictions. Intra-operative blood loss exceeding 10% EBV was associated with OT > 250 min, primary surgery, serous histology, performance status 0, R2 resection and surgical complexity score > 4. Certain sub-procedures including large bowel resection, stoma formation, ileocecal resection/right hemicolectomy, mesenteric resection, bladder and upper abdominal peritonectomy demonstrated clear associations with an elevated interventional risk. Our findings emphasize the importance of obtaining a rough estimate of OT in advance for precise prediction of blood requirements.

## 1. Introduction

Institutional transfusion protocols are not universal, and a variety of transfusion policies may exist across participating institutions. The recently published European Society of Gynaecological Oncology (ESGO) guidelines for the peri-operative management of advanced epithelial ovarian cancer (EOC) patients clearly stated the absence of a well-defined threshold for BT in cytoreductive surgery [1]. Indeed, since many patients will require adjuvant chemotherapy, more liberal transfusion thresholds may be considered. Equally, the BT requirements have long been debated because they are not without risk [2]. Intra-operative decisions are critical in the surgical management of advanced EOC. In such a dynamic theatre environment, harmonic coordination amongst team members including surgeons, anaesthetists and theatre staff can lead to procedural success. Surgical decision-making incorporates situational awareness and is affected by the dynamics of human factors [3]. While clinical judgments are enhanced over years of training, risk prediction tools still provide the opportunity to augment decision-making. Nevertheless, traditional models fall short to address the complex interactions amongst discrete prediction variables. Machine Learning-based prediction tools can potentially rise above the peak of inflated expectations [4]. Identifying the optimal BT threshold for patients undergoing cytoreduction for advanced EOC becomes crucial, considering the potential risks associated with the surgery.

We hypothesized that pre-operative and intra-operative risk factors for planned cytoreductive procedures can help predict anticipated blood requirements, use of haemostatic sealants and, subsequently anaesthetic interventions in the peri-operative period. We aimed to devise an ML-driven algorithm for transfusion prediction that could trigger a BT communication alert based on anticipated intra-operative events.

## 2. Methods

We analyzed prospectively collected data from 560 patients with advanced epithelial ovarian cancer (EOC) who underwent cytoreductive surgery at a UK tertiary center between January 2014 and December 2019. The internally developed database was integrated with the Electronic Health Records (EHRs). Institutional ethics approval was secured from Leeds Teaching Hospitals Trust (MO20/133163/18.06.20), and informed written consent was obtained for the study. The research was registered in the UMIN/CTR Trial Registry (UMIN000049480). The surgical approach was determined during the weekly central Multidisciplinary Meeting (MDT) before the patient’s pre-operative review. The cohort characteristics have been previously detailed [5]. Exclusion criteria encompassed patients aged < 18 years old, those with non-epithelial histology, women with synchronous primary malignancy, and those undergoing secondary cytoreduction for recurrent disease. Additionally, women with pre-operative anaemia (defined as a haemoglobin level < 120 g/dL) and confirmed coagulopathy (assessed through medical history or routine coagulation results) were excluded. The intra-operative fluid protocol consisted of both maintenance therapy (1 mL/kg/h of crystalloid) and replacement of blood loss (1 mL of crystalloid for each mL of directly measured blood loss). If necessary, vasoactive agents were titrated to obtain a mean blood pressure > 65 mmHg. The EBL was estimated by subtracting the intra-operative fluid losses suctioned through suctioned drains into the surgical canister from the total blood volume contained in the surgical canister at the end of the procedure and empirically adding the blood loss on the required surgical gauzes.

Pre-operative features included age, Eastern Co-operative Oncology Group (ECOG) performance status (PS), histology type (serous and non-serous), grade (low and high), stage (FIGO 3 or 4), as well as pre-treatment and pre-surgery CA-125. Operative/tumor factors comprised timing of surgery (Primary Debulking Surgery (PDS) or Interval Debulking Surgery (IDS), operative time (OT), estimated blood loss (EBL), site and size of the main disease bulk, Peritoneal Carcinomatosis Index (PCI) [6], intra-operative mapping of ovarian cancer (IMO) [7], Surgical Complexity Score (SCS) [8], and residual disease. Residual disease was assessed through a comprehensive visual examination of all abdominopelvic regions. Human factors considered were the age of the consultant surgeon and years of experience. Surgical sub-procedures were documented using the adapted version of the structured ESGO Ovarian Cancer Operative Report aligning with quality indicators (Qis) for ovarian cancer surgery [9]. This report provided detailed information on tumor dissemination patterns (https://guidelines.esgo.org/, accessed on 10 June 2023). Descriptive statistics were presented by frequency and percentages for binary and categorical variables and means with standard deviation (SD) or medians (with lower or upper quartiles for continuous variables). Continuous variables were analyzed using the *t*-test, categorical variables with the Chi-square test, and binary variables with Fischer’s exact tests. Statistical significance was set established at *p* < 0.05. Overall survival analysis employed Kaplan–Meier and Mantel–Cox log-rank tests for comparison.

The eXtreme Gradient Boosting (XGBoost) algorithm was employed to model the features [10]. It enhances the performance of weak learning algorithms by combining their generated hypotheses into a singular aggregated hypothesis. To optimize model performance, we examined the combined impact of eight parameters by evaluating a grid comprising 7680 combinations of values through Scikit-learn’s GridSearchCV function. In the absence of standardized definitions for intra-operative bleeding, the study consisted of a concept phase. During this phase, we calculated the estimated blood volume (EBV) using the formula EBV = weight × 80 and established as the threshold for individual intervention [11]. This 10% EBV threshold was employed to frame a binary prediction classification problem. Based on known EBL, two subgroups were identified, and the dataset was partitioned into training and test cohorts (in a 70%:30% ratio). No significant differences (*p* > 0.20) were observed between the two cohorts for all variables. Five-fold stratified cross-validation (CV) was conducted, creating stratified folds to ensure equal distribution between folds above and below the threshold. The CV was used to decrease both variance and bias, resulting in the formation and evaluation of 38,400 models. Model performance was assessed by measuring the total area under the receiver-operating curve (AUC). Receiver operating characteristic (ROC) and Precision-Recall curves along with state-of-the-art scores for performance metrics. The artificial intelligence SHapley Additive exPlanations (SHAP) framework was employed to explain the cohort-level risk estimates, and to define novel surgical risk phenotypes [12]. This methodology enhances interpretability by explaining the contribution of each feature to the model’s overall prediction. To illustrate the value of the model prediction, we generated (a) SHAP summary plots for the global (cohort) explanation of the results; (b) SHAP dependence plots highlighting critical risk features relevant to the prediction. The analyses were conducted using the Python’s SciPy library (version 2.7) (Python Software Foundation) Python Language Reference, version 2.7, available at http://www.python.org, accessed on 10 June 2023.

## 3. Results

The study initially enrolled 560 EOC patients. Exclusions were made for 117 patients with pre-operative anaemia, and 26 patients with abnormal coagulation systems, respectively. Four patients were further excluded: one patient with a synchronous primary malignancy and three patients undergoing emergency cytoreduction with no intention to treat. An additional 10 patients were omitted due to incomplete data. Ultimately, 403 EOC patients participated in the final analysis (Table 1, Figure 1).

Follow-up extended until April 2022. The mean age for the entire cohort was 65.31 + 11.2 years, and this was comparable between the train and test sets (63.2 ± 11, 64.6 ± 12, respectively (p:NS). Histology, PS, tumour grade and stage, timing of surgery, residual disease, SCS, IMO score, PCI, pre-surgery Ca 125 and occurrence of ascites demonstrated no significant difference between the training and test sets (p:NS). Except for pre-treatment Ca 125 and age, all variables were statistically significant between the < and >10% EBV groups. Details of surgical sub-procedures have been previously published [13].

The model performance for the above threshold prediction was moderate-to-high (AUC 0.82, 95% CI 0.78–0.86; AP 0.72, 95% CI 0.67–0.76) (Figure 2). To promote reproducibility, the optimal set of model parameters is available upon reasonable request.

The importance of features in this model based on SHAP values is shown in Figure 3. The order of the features reflects their weighted importance, i.e., the sum of the SHAP value magnitudes across all the samples. Each point on the summary plot is a Shapley value for a feature. The position on the y-axis is determined by the feature and on the x-axis by the Shapley value. The color represents the value of the feature from low (blue =< 10% EBV) to high (red => 10% EBV). The top feature commonly shared between both interrogators was operative time (OT). The top-five list of important features included pre-treatment CA125, logCa125/PCI, a marker of biological tumor aggressiveness, followed by the SCS and the size of the largest disease bulk. Long tails were observed mainly for operative time and size of the largest tumor bulk, which suggests these features can be equally important for specific if not all patients (local explainability).

The SHAP dependence plots reveal the impact of each feature on the prediction by plotting the value of the feature on the x-axis and the SHAP feature value on the y-axis. While Figure A1A fails to show a clear inflection point, inflection points are demonstrated for high grade (Figure A1B), PS > 1 (Figure A1C) and serous histology (Figure A1D). When overall SHAP values are positive, the likelihood for BT intervention is higher. For OT < 150 min, the overall SHAP values are negative, which makes BT intervention unlikely (Figure A2A). Then, there is no clear inflection point for OT ranging between 150 min and 250 min. Subsequently, for OT > 250 min, it becomes more likely to administer BT. If PCI is selected as a feature to determine its impact, an increasing PCI up to eight results in a lower likelihood for BT (Figure A2D). Similarly, the inflection points for IMO score and SCS are four and three, respectively. Clearly, delayed surgery and R0 resection do not favor BT intervention (Figure A2B,F). From the list of surgical sub-procedures (Table 2) clearly reflecting radical or ultra-radical cytoreductive efforts, an intra-operative blood loss of at least 10% EBV was associated with the performance of (A) stoma formation (Figure A3A), bladder peritonectomy (Figure A3B), para-aortic lymphadenectomy (Figure A3C), ileo-caecal resection/right hemicolectomy (Figure A3D), mesenteric resection (Figure A3E), upper abdominal peritonectomy (Figure A3F), and large bowel resection (Figure A3G).

The five-year OS for the entire cohort was 61 months (95% CI 58–63). The median OS was 60 months for the <10% EBV group (95% CI 58–61) and 61 months (95% CI 60–63) for the >10% EBV group (p:NS). In the predicted >10% EBV group, only 10% (*n* = 23) received BT (Figure 4).

## 4. Discussion

One of the recently revised Quality Indicators (QI7) among the 10 designated by ESGO emphasizes the significance of sufficient anesthesiology and peri-operative care to ensure optimal surgical results in advanced EOC cytoreduction [9]. This process indicator is dedicated to not only minimizing surgical morbidity but also enhancing the efficiency of facilities and personnel for effective complication management. The guideline acknowledges the absence of a defined threshold for intra-operative BT. We seek to address this issue by devising and elucidating an ensemble ML algorithm for predicting intra-operative risk during advanced EOC cytoreduction (Figure 5). Equally, we assessed the impact of independent patient- disease- and operation-specific features on intra-operative blood loss.

While advocating for standardized definitions of intra-operative bleeding in surgical practice, we conceptually established a threshold for potential personalized intervention, irrespective of BT policies. This straightforward tool, based on EBV and EBL, has the potential to improve the language and communication surrounding blood loss. Integrating percentage EBV triggers into the surgical safety list may reduce the requirement for additional interventions. Such mental preparation can significantly contribute to surgical success [14]. To our knowledge, this represents the initial effort to identify an objective threshold that could prompt interventions inclusive of BT or use of haemostatic agents during EOC cytoreductive surgery.

The study explains how EBL can be divided into a single contributing component, which is the surgical time. As such, the main study finding is that precise prediction of blood requirements or strategies to address haemostasis is not possible unless a rough estimate of OT is known in advance. An inflection point was not identified, but OT > 250 min clearly increases the risk for BT administration.

The inter-related events of blood transfusion and operating room procedures emphasize the need for quality assurance in all intra-operative events, which should be a focus of quality assurance efforts in the future [15]. Interestingly, the top-five list of important features included not just markers of surgical aggressiveness but also inherent tumor biology, such as the pre-treatment Ca125 or the Ca125/PCI ratio. This information is clinically relevant, and can be used in studies investigating haemostatic agents [11]. Previous studies identified age and low baseline hemoglobin level as risk factors for transfusion in this population [16,17]. Malignant ascites > 500 mL can be a risk factor as they imply increased fluid demands and substantial alterations in circulatory blood flow during cancer surgery [2], but this was not shown in our study. Lately, we read with interest the development of a new tool that can predict the need for BT during or after primary debulking surgery in EOC patients. The tool, called the “BLOODS score” was developed to help identify patients who may benefit from pre-operative planning and blood-saving techniques [18]. The score was created using data from a similar practice setting. In a cohort totaling 1289 patients, and for a median estimated blood loss of 650 cc, the seven features associated with BT included the following: the American Society of Anesthesiologists physical score of 3 or higher, the pre-operative serum albumin level of 3.5 g/dL or lower, presence of ascites, pre-operative CA-125 level of 600 U/mL or higher, carcinomatosis, moderate or high SCS compared to low SCS, (h), pre-operative creatinine level greater than 0.90 mg/dL, and upper abdominal disease of 1 cm or greater. The features were assigned a score of 1 if present. The sum of the BLOODS score corresponded to the need for BT. The authors claim that a BLOODS score of 3 or higher is estimated to identify 80% of patients who require a BT, including 85% of patients who need a transfusion intra-operatively and 78% who need one post-operatively. Notably, the rate of full clearance was just above 50%. In our study, we clearly demonstrated that the risk for BT is associated with primary surgery- owing to a complete cytoreduction rate of 67.3%- too. It is likely that with expansion of our surgical cohort, we may encounter similar results in the future.

At our tertiary center, patient selection for cytoreductive surgery is based on pre-operative evaluation using contrast-enhanced cross-sectional imaging. However, the most crucial aspect of the selection process is the rigorous discussion and decision-making carried out by the Multidisciplinary Team (MDT). Surgeons are requested to estimate the likelihood of extensive cytoreduction and pre-operatively anticipate the input values for several intra-operative variables. We are not surprised why OT was the single most important contributor to the threshold risk for potential intervention. In theory, the duration of surgery can vary depending on the extent of disease spread, the surgical effort, and the experience of the surgical team in addition to the patient’s overall health.

Anticipating the procedural timing is a crucial aspect of organizing theatrical events, requiring an understanding of the surgical efforts needed at the start of the performance. This foresight can streamline scheduling and help predict resource requirements. Efficient utilization of the operating room depends on precise estimates of surgical control time. With the rapid rise in the number of EOC diagnosis, there is a growing need for efficient management in theatre time. Anecdotal evidence suggests that surgeons grossly underestimate predicted surgical time [19].

Herein, we provided clear evidence of several surgical sub-procedures, indirectly linked to procedural time, which are associated with higher risk for intra-operative intervention. Nevertheless, an extensive surgical approach is associated with increased blood loss and potentially higher peri-operative complications [20]. As studies continue to share knowledge about the optimal duration of surgical procedures [21], another study is currently underway to assign optimal procedural times based on knowledge retrieved from operative notes using natural language processing (data not shown).

This focus holds potential implications including enhancing information delivery to surgeons to promote meticulous planning for complication reduction. Additionally, it could contribute to improved patient counseling and discussions regarding the scope of surgery or alternative therapeutic options. Existing studies focused on developing models to identify patients at the lowest risk, potentially requiring less pre-operative BT preparation [22,23]. Utilizing this information, tailored EOC-specific institutional guidelines can be formulated to anticipate crossmatch and transfusion requirements, fostering judicious use of blood products, and facilitating cost savings. Owing to the high prevalence of BT in this population, the insights derived can add significant value to accurate surgical planning and the pre-operative optimization of potentially modifiable factors.

We acknowledge that initial intra-operative anaesthetic interventions do not always involve transfusions of blood products. Goal-directed haemodynamic algorithms are routinely used to optimize stroke volume and provide better systemic stability and reduced need for fresh-frozen plasma [24]. Efforts to promote sensible use of intra-operative BT also include acute normovolemic haemodilution without increasing peri-operative complications [25]. Other strategies for “bloodless” surgery have been recently described [26,27]. There is insufficient evidence to recommend the routine use of tranexamic acid for reducing blood loss in women undergoing cytoreductive surgery for advanced EOC, as only limited data are available from a single, low quality RCT [28]. Optimizing intra-operative fluid management may improve short-term patient outcomes [29]. Notably, the intra-operative BT requirements are significantly lower in prehabilitation groups of EOC candidates to cytoreductive surgery [28]. The impact of BT on EOC survival has been uncertain until the ancillary analysis of the EORTC 55971 phase III trial, which concluded that BT does not negatively impact progression-free survival or overall survival. Nevertheless, it was associated with increased peri-operative morbidity without improvements in quality of life [25]. The case for overall survival was presented in our study albeit only 10% of patients in the >10% EBV group (rule of thumb) received BT peri-operatively. Intra-operative bleeding is the most relevant finding to surgical complications. Surgical time may not be related to complications, which are significantly related to delayed chemotherapy [30].

The strengths of this study included a large, population-based database inclusive of patient- and cancer-specific, but also surgical features. Causality could not be inferred in variables predicting the primary outcome. The use of an ML-based algorithm reliably predicted the risk for potential intervention. We used explainability AI to unveil the “black box” of ML, which is critical to overcome any resilience for clinical implementation. In gynaecological oncology, translatability of models is important when it comes to endpoint selections, due to the significant variability in model development [31]. Should another threshold be used, the results would have been different. Indeed, we tested the 20%EBV cut off, but the results were poor due to group imbalance. Despite the rigorous five-fold internal validation, this single-center study design cannot provide real-world risk estimation for experienced centers. The top predictor could only be extracted from the operative details, which limits pre-operative counseling. Prospective evaluation may allow for model integration into EHR.

## 5. Conclusions

We identified the primary risk factor for intra-operative bleeding in EOC patients undergoing cytoreductive surgery that necessitates BT or the use of surgical adjuncts. When patients exhibit no signs of anaemia and retain an intact anticoagulation system, the occurrence of intra-operative blood loss exceeding 10% of EBV was primarily associated with OT followed by pre-treatment CA125, and a suboptimal surgical outcome. The ROC curves indicated that by integrating these features into a single ML-based prediction system, specifically XGBoost, reliable predictions of intra-operative blood loss and associated requirements were demonstrated. This valuable information not only has the potential to prevent serious post-operative mortality and morbidity but also aids in avoiding unnecessary depletion of viral blood resources.

## Figures and Tables

**Figure 1 diagnostics-14-00094-f001:**
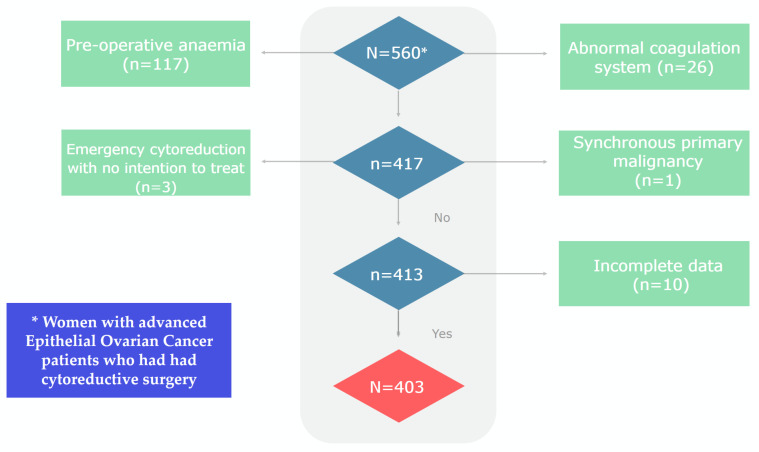
Flowchart of the study cohort.

**Figure 2 diagnostics-14-00094-f002:**
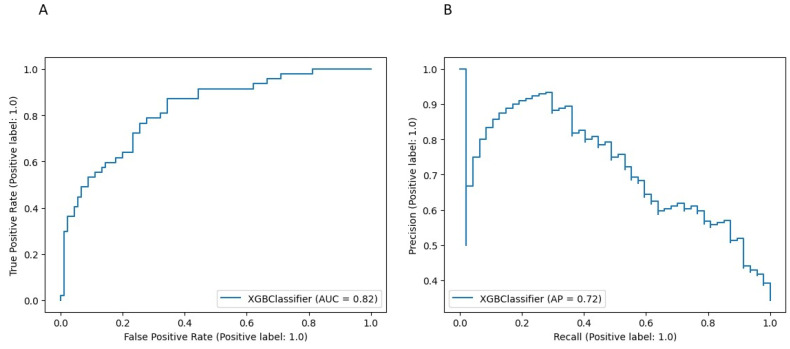
Performance of the XGBoost model for the risk prediction of blood transfusion at cytoreductive surgery (**A**) Receiver Operator Characteristic (ROC) curve. (**B**) Precision Recall (PR) curve.

**Figure 3 diagnostics-14-00094-f003:**
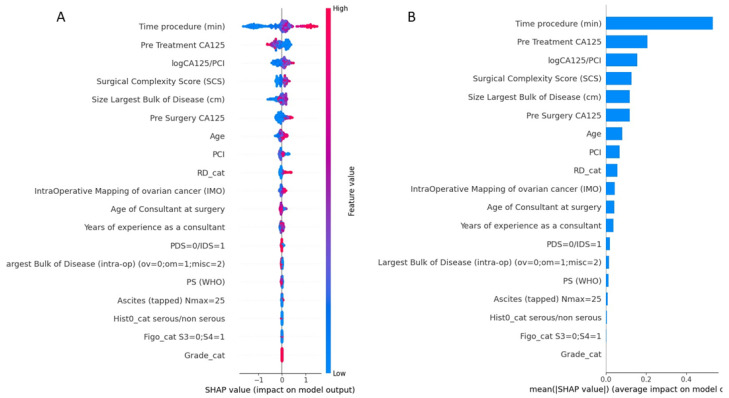
(**A**) Summary plot showing a set of feature distribution beeswarm plots for global (threshold) explainability of 10% EBV threshold prediction. (**B**) Feature importance bar plot of their SHAP values. PCI, Peritoneal Carcinomatosis Index; RD, Residual Disease; PDS, Primary Debulking surgery; IDS, Interval Debulking Surgery; PS, Performance Status.

**Figure 4 diagnostics-14-00094-f004:**
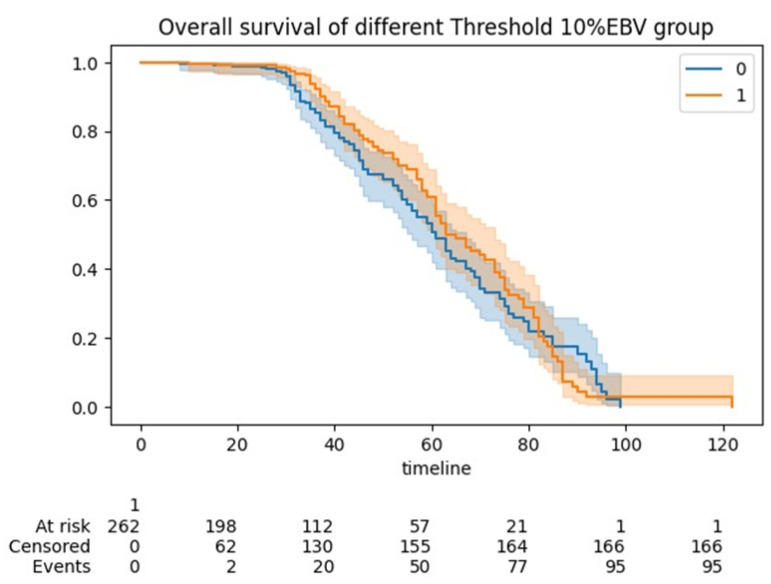
Kaplan Meier (KM) curve showing overall survivals between the <10% EBV and >10% EBV threshold cohorts of women undergoing cytoreduction for advanced epithelial ovarian cancer. No statistical significance was demonstrated.

**Figure 5 diagnostics-14-00094-f005:**
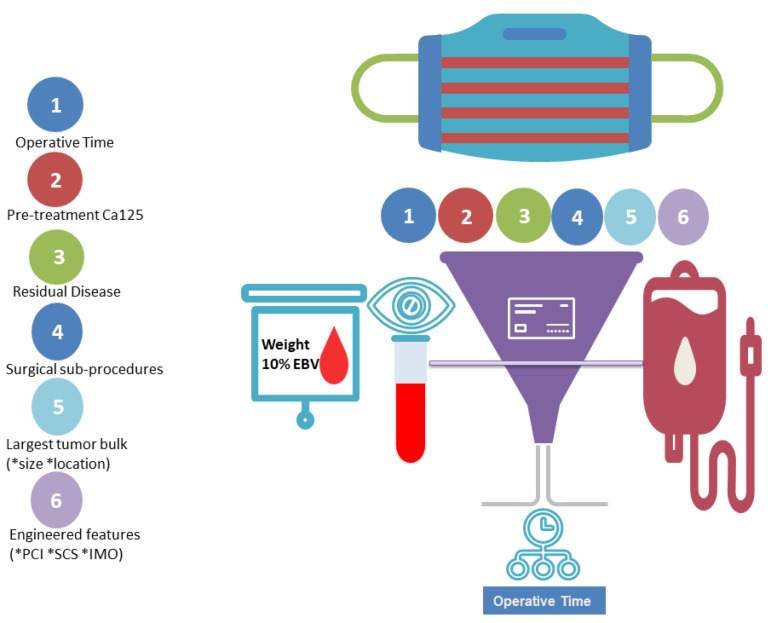
Schematic representation of our study. According to our concept, ML-based feature selection identified operative time out of an exhaustive list of patient, disease and operation-specific features as the top feature for the risk prediction of blood transfusion at cytoreductive surgery.

**Table 1 diagnostics-14-00094-t001:** Cohort statistics.

		Overall	Training Set	Testing Set	*p* Value (Training)	Group < 10% EBV	Group > 10% EBV	*p* Value (Threshold 10% EBV)
Histo_cat ^1^	0	489 (87.32)	394 (87.95)	95 (84.82)	0.465	264 (88.59)	137 (87.82)	0.929
	1	71 (12.68)	54 (12.05)	17 (15.18)	0.465	34 (11.41)	19 (12.18)	0.929
Grade_cat	0	56 (10.0)	46 (10.27)	10 (8.93)	0.805	20 (6.71)	22 (14.1)	0.016
	1	504 (90.0)	402 (89.73)	102 (91.07)	0.805	278 (93.29)	134 (85.9)	0.016
Figo_cat ^2^	0	406 (72.5)	322 (71.88)	84 (75.0)	0.586	216 (72.48)	113 (72.44)	1
	1	154 (27.5)	126 (28.12)	28 (25.0)	0.586	82 (27.52)	43 (27.56)	1
PS (WHO)	0	266 (47.5)	205 (45.76)	61 (54.46)	0.099	137 (45.97)	76 (48.72)	0.048
	1	208 (37.14)	175 (39.06)	33 (29.46)	0.099	111 (37.25)	65 (41.67)	0.048
	2	67 (11.96)	56 (12.5)	11 (9.82)	0.099	43 (14.43)	9 (5.77)	0.048
	3	17 (3.04)	11 (2.46)	6 (5.36)	0.099	7 (2.35)	5 (3.21)	0.048
	4	2 (0.36)	1 (0.22)	1 (0.89)	0.099	0 (0.0)	1 (0.64)	0.048
PDS = 0/ IDS = 1	0	172 (30.71)	132 (29.46)	40 (35.71)	0.243	72 (24.16)	61 (39.1)	0.001
	1	388 (69.29)	316 (70.54)	72 (64.29)	0.243	226 (75.84)	95 (60.9)	0.001
RD_cat	0	369 (65.89)	294 (65.62)	75 (66.96)	0.168	209 (70.13)	93 (59.62)	0.051
	1	130 (23.21)	100 (22.32)	30 (26.79)	0.168	64 (21.48)	41 (26.28)	0.051
	2	61 (10.89)	54 (12.05)	7 (6.25)	0.168	25 (8.39)	22 (14.1)	0.051
Ascites ^3^	0	414 (73.93)	335 (74.78)	79 (70.54)	0.427	234 (78.52)	103 (66.03)	0.005
	1	146 (26.07)	113 (25.22)	33 (29.46)	0.427	64 (21.48)	53 (33.97)	0.005
Largest Bulk of Disease Location ^4^	PA node	1 (0.18)	1 (0.22)	0 (0.0)	0.023	1 (0.34)	0 (0.0)	0.446
	POD	1 (0.18)	0 (0.0)	1 (0.89)	0.023	1 (0.34)	0 (0.0)	0.446
	caecum	1 (0.18)	0 (0.0)	1 (0.89)	0.023	1 (0.34)	0 (0.0)	0.446
	mesentery	2 (0.36)	2 (0.45)	0 (0.0)	0.023	0 (0.0)	2 (1.28)	0.446
	omentum	249 (44.46)	204 (45.54)	45 (40.18)	0.023	138 (46.31)	71 (45.51)	0.446
	ovary	300 (53.57)	238 (53.12)	62 (55.36)	0.023	154 (51.68)	82 (52.56)	0.446
	peritoneum	1 (0.18)	1 (0.22)	0 (0.0)	0.023	0 (0.0)	1 (0.64)	0.446
	rectum	1 (0.18)	0 (0.0)	1 (0.89)	0.023	1 (0.34)	0 (0.0)	0.446
	sigmoid	1 (0.18)	0 (0.0)	1 (0.89)	0.023			0.446
	umbilicus	1 (0.18)	0 (0.0)	1 (0.89)	0.023	1 (0.34)	0 (0.0)	0.446
Age		63.51 ± 11.22	63.23 ± 11.06	64.64 ± 11.82	0.252	63.53 ± 11.24	62.78 ± 11.27	0.502
Consultant age ^5^		49.13 ± 6.03	49.1 ± 6.07	49.25 ± 5.91	0.815	49.96 ± 6.09	48.43 ± 6.11	0.011
Years ^6^		9.65 ± 5.33	9.65 ± 5.36	9.68 ± 5.22	0.952	10.03 ± 5.31	9.29 ± 5.47	0.165
SCS ^7^		3.8 ± 2.11	3.82 ± 2.06	3.71 ± 2.31	0.648	3.34 ± 1.83	4.73 ± 2.51	<0.001
Time procedure ^8^		170.39 ± 77.55	172.98 ± 76.53	160.04 ± 81.03	0.129	147.84 ± 55.24	215.1 ± 97.06	<0.001
Pre Treatment CA125		1516.14 ± 2711.14	1582.85 ± 2769.98	1249.29 ± 2455.18	0.212	1689.69 ± 3189.63	1420.7 ± 2071.05	0.279
Pre Surgery CA125		410.46 ± 1175.43	411.43 ± 944.52	406.56 ± 1833.3	0.978	360.46 ± 1280.81	614.46 ± 1298.66	0.048
logCA125/ PCI		0.41 ± 0.36	0.4 ± 0.35	0.42 ± 0.4	0.756	0.41 ± 0.35	0.4 ± 0.41	0.657
IMO score ^9^		4.92 ± 1.97	4.98 ± 1.99	4.7 ± 1.89	0.158	4.57 ± 1.86	5.6 ± 2.11	<0.001
PCI		7.37 ± 4.47	7.48 ± 4.51	6.92 ± 4.31	0.225	6.78 ± 4.08	8.79 ± 5.16	<0.001
Largest Bulk (cm)		8.89 ± 5.61	9.13 ± 5.69	7.96 ± 5.23	0.039	8.29 ± 5.64	9.98 ± 5.49	0.002

^1^ serous/non serous, ^2^ S3 = 0; S4 = 1, ^3^ (tapped) Nmax = 25, ^4^ (intra-op) (ov = 0; om = 1; misc = 2), ^5^ Age of Consultant at surgery, ^6^ Years of experience as a consultant, ^7^ Surgical Complexity Score, ^8^ in minutes, ^9^ IntraOperative Mapping of ovarian cancer, PS: performance status, PDS: primary debulking surgery, IDS: interval debulking surgery, RD: residual disease, PA: para-aortic, POD: pouch of Douglas, SCS: surgical complexity score, IMO: intra-operative mapping of ovarian cancer, PCI: peritoneal carcinomatosis index.

**Table 2 diagnostics-14-00094-t002:** Surgical sub-procedures’ statistics for the whole dataset broken down by CC0/non-CC0 patients. Chi-Square Test of Independence was used. Statistical analysis was performed using Python’s SciPy library. Values are n (%).

		Overall (n = 560)	CC0 (n = 368)	Non-CC0 (n = 192)	*p*-Value
Stoma Formation	0	509 (90.89)	334 (90.76)	175 (91.15)	1
	1	51 (9.11)	34 (9.24)	17 (8.85)	1
Bladder Peritonectomy	0	358 (63.93)	217 (58.97)	141 (73.44)	0.001
	1	202 (36.07)	151 (41.03)	51 (26.56)	0.001
Para-aortic node dissection	0	381 (68.04)	221 (60.05)	160 (83.33)	<0.001
	1	179 (31.96)	147 (39.95)	32 (16.67)	<0.001
Ileo-Caecal Resection/ Right Hemicolectomy	0	539 (96.25)	352 (95.65)	187 (97.4)	0.426
	1	21 (3.75)	16 (4.35)	5 (2.6)	0.426
Mesenteric Resection	0	427 (76.25)	269 (73.1)	158 (82.29)	0.02
	1	133 (23.75)	99 (26.9)	34 (17.71)	0.02
Upper Abdominal Peritonectomy	0	481 (85.89)	296 (80.43)	185 (96.35)	<0.001
	1	79 (14.11)	72 (19.57)	7 (3.65)	<0.001
Large Bowel Resection	0	496 (88.57)	323 (87.77)	173 (90.1)	0.494
	1	64 (11.43)	45 (12.23)	19 (9.9)	0.494
Pelvic node dissection	0	414 (73.93)	242 (65.76)	172 (89.58)	<0.001
	1	146 (26.07)	126 (34.24)	20 (10.42)	<0.001

## Data Availability

The data presented in this study are available on request from the corresponding author.

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
