# Peer review of "Explaining the Elusive Nature of a Well-Defined Threshold for Blood Transfusion in Advanced Epithelial Ovarian Cancer Cytoreductive Surgery"

_diagnostics, 2023, doi:10.3390/diagnostics14010094_

Round 1
Reviewer 1 Report
Comments and Suggestions for Authors
The manuscript appears interesting because it addresses a topic that is certainly important for the clinical history of patients affected by ovarian cancer. I do not believe that the application of this algorithm can have an important impact on the patient's clinical history as the management of the oncology patient, as described in this study, is part of the good clinical practice of every gynecological oncology centre, so I do not find any great scientific relevance.
Author Response
Thank you for your comments. The scope of this manuscript is not to challenge the clinical roadmap of advanced ovarian cancer patients. There is a knowledge gap identified in the recently published guidelines on the peri-operative surgical management of advanced stage ovarian cancer patients; the lack of a clearly defined threshold for intra-operative blood transfusion at cytoreduction. In fact, that reflects the QI 7 among the 10 designated ESGO published QIs. This process indicator emphasizes the significance of sufficient anaesthetic and peri-operative care to ensure optimal surgical results in advanced EOC cytoreduction. It is dedicated to not only minimizing surgical morbidity but also enhancing the efficiency of facilities and personnel for effective management of complications. The manuscript addresses this issue by identification of the primary risk factor for intra-operative bleeding and emphasizing on the importance of obtaining a rough estimate of operative time in advance for precise prediction of blood requirements. This information can trigger a communication alert to not only prevent serious post-operative mortality and morbidity but also avoiding unnecessary depletion of viral blood resources. The first and second paragraph in the discussion section clearly address the Reviewer's concerns in addition to all the highlighted in red rewritten paragraphs.
Reviewer 2 Report
Comments and Suggestions for Authors
I have read the present manuscript with interest. The original idea of the study is very interesting and the methodology used is adequate to extract safe conclusions. However, before publication some issues have to be solved:
1. Please add a flowchart in the results section presenting schematically the first paragraph.
2. I believe that figures 3-5 is better to be cited in Supplementary Materials.
3. Please seperate the extensive paragraphs and add paragraphs at line 61, 74, 172,198, 213, 247
Comments on the Quality of English LanguageMinor typping errors
Author Response
1) A flowchart of the study cohort selection was created and presented as Figure 1.
2)The Figures 3-5 were cited as supplementary materials as A1-3.
3) Extensive paragraphs were separated and new paragraphs were added accordingly to the best of our effort.
Reviewer 3 Report
Comments and Suggestions for Authors
Laios and collaborators present very interesting research on the identification of preoperative and operative parameters with the most effective predictive value in anticipating the need for blood transfusion during cytoreductive surgery for advanced ovarian cancer.
The work is original, well presented and well written. The results are clearly presented and honestly analyzed.
Author Response
Thank you for your positive feedback that values our work. Much appreciated.